# Tumor-like Lesions in Primary Angiitis of the Central Nervous System: The Role of Magnetic Resonance Imaging in Differential Diagnosis

**DOI:** 10.3390/diagnostics14060618

**Published:** 2024-03-14

**Authors:** Marialuisa Zedde, Manuela Napoli, Claudio Moratti, Claudio Pavone, Lara Bonacini, Giovanna Di Cecco, Serena D’Aniello, Ilaria Grisendi, Federica Assenza, Grégoire Boulouis, Thanh N. Nguyen, Franco Valzania, Rosario Pascarella

**Affiliations:** 1Neurology Unit, Stroke Unit, Azienda Unità Sanitaria Locale-IRCCS di Reggio Emilia, Viale Risorgimento 80, 42123 Reggio Emilia, Italy; grisendi.ilaria@ausl.re.it (I.G.); assenza.federica@ausl.re.it (F.A.); valzania.franco@ausl.re.it (F.V.); 2Neuroradiology Unit, Azienda Unità Sanitaria Locale-IRCCS di Reggio Emilia, Viale Risorgimento 80, 42123 Reggio Emilia, Italy; napoli.manuela@ausl.re.it (M.N.); moratti.claudio@ausl.re.it (C.M.); bonacini.lara@ausl.re.it (L.B.); daniello.serena@ausl.re.it (S.D.); pascarella.rosario@ausl.re.it (R.P.); 3Departments of Diagnostic and Interventional Neuroradiology, University Hospital Tours, Centre Val de Loire Region, 37100 Tours, France; gregoireboulouis@gmail.com; 4Neurology, Boston University School of Medicine, Boston, MA 02118, USA; thanh.nguyen@bmc.org; 5Radiology, Boston Medical Center, Department of Radiology, Boston, MA 02118, USA

**Keywords:** PACNS, tumor-like lesion, mass-like lesion, tumefactive lesion, PCNSL, glioma, HGG, TDL, sarcoidosis, neurotoxoplasmosis, MRI

## Abstract

Primary Angiitis of the Central Nervous System (PACNS) is a rare disease and its diagnosis is a challenge for several reasons, including the lack of specificity of the main findings highlighted in the current diagnostic criteria. Among the neuroimaging pattern of PACNS, a tumefactive form (t-PACNS) is a rare subtype and its differential diagnosis mainly relies on neuroimaging. Tumor-like mass lesions in the brain are a heterogeneous category including tumors (in particular, primary brain tumors such as glial tumors and lymphoma), inflammatory (e.g., t-PACNS, tumefactive demyelinating lesions, and neurosarcoidosis), and infectious diseases (e.g., neurotoxoplasmosis). In this review, the main features of t-PACNS are addressed and the main differential diagnoses from a neuroimaging perspective (mainly Magnetic Resonance Imaging—MRI—techniques) are described, including conventional and advanced MRI.

## 1. Introduction

Primary Angiitis of the Central Nervous System (PACNS) is a rare disease involving the arteries of the brain, spinal cord, and leptomeninges without other systemic localization. It is characterized by the pathological finding of a transmural inflammatory infiltrate within the vessel wall [1]. In routine clinical practice, it is possible to have a pathological confirmation of the diagnosis only in a minority of cases, depending on the size of the involved vessels and on the site of brain parenchymal damage. The subtype of disease involving large vessels (LV-PACNS) is out of reach of a biopsy, but the subtype involving small vessels (SV-PACNS) cannot be diagnosed without pathological confirmation [2], according to current diagnostic criteria [3,4]. These criteria have several limitations and a strong validation was not performed before their proposal in 1988 [3] and 2009 [4]. The diagnosis of PACNS is still a challenge in most cases and, as outlined in the recently published guidelines of the European Stroke Organization (ESO) on PACNS [5], there is wide heterogeneity and variability in performing neuroimaging studies, interpreting and reporting the findings, adhering to the diagnostic criteria [3,4], and registering their outcomes. This lack of a systematic method, of particular impact in reporting neuroimaging findings of brain parenchyma and vessels, is one of the main factors limiting the possibility to correlate specific neuroimaging phenotypes to histopathologic subtypes and natural history. Moreover, this missing information makes each neuroimaging finding non-specific for PACNS diagnosis, making the differential diagnosis from other diseases difficult. Furthermore, each of the diagnostic cornerstones of PACNS (cerebrospinal fluid analysis, neuroimaging of brain parenchyma, neuroimaging of cerebral vessels, and even pathology) has defined its subset of differential diagnoses over time. An integrated neurological and neuroradiological approach, starting from a background and expertise dedicated to cerebrovascular diseases, could help to achieve the following:-Collect prospective data of adequate quality to validate some findings against an updated subset of differential diagnoses;-Define clinical and neuroradiological predictors of PACNS diagnosis;-Form the basis for designing treatment trials.

Beginning with the identification and description of neuroradiological patterns of brain involvement probably represents the most promising and reasonable approach to diagnose PACNS. The aim of this narrative review is to propose some decision-making points in the differential diagnosis of PACNS, based on advanced neuroimaging study, in a particular and challenging neuroradiological phenotype, namely, the tumefactive or pseudotumoral subtype.

## 2. Tumor-like Lesions in PACNS

In the recent ESO Guidelines on PACNS [5], the predictive value of some predefined neuroimaging patterns of parenchymal involvement on brain MRI was investigated on the basis of the available literature. The working group proposed the following neuroimaging patterns (Table 1) and the literature review proposed by the guidelines [5] collected data on 660 PACNS patients.

Although underreporting is probable for most of these patterns, the pseudotumoral pattern emerges as extremely rare, even in single-center case series covering several decades. Furthermore, the differential diagnosis of tumefactive lesions is complex from a neuroradiological point of view and includes inflammatory, demyelinating, infectious, vascular, and neoplastic diseases, whose treatment is often opposite to that required in the diagnostic hypothesis of PACNS. Furthermore, some therapies are often started before acquiring a sample of brain tissue, on the basis of elements supporting one diagnostic hypothesis or other, mainly based on neuroradiological findings, which can sometimes lead to an inconclusive histopathology. 

The use of a standardized diagnostic approach and the valorization of neuroimaging findings at decision-making points, including advanced neuroimaging techniques, such as perfusion MRI, could help to make fewer discretionary decisions compared to the management of the individual case. Another issue is that the neuroimaging patterns outlined above are not mutually exclusive, but can coexist, both simultaneously in the same investigation and in a metachronous form in the evolution of the neuroimaging of the same patient.

### 2.1. Definition

Several diseases are known to have tumefactive patterns in neuroimaging studies among their manifestations. A unique definition of tumefactive lesion does not exist, apart from the space-occupying feature as suggested by the name. Indeed, some papers reported as a macrocategory the umbrella term of “tumefactive inflammation” [8], grouping together different neurological diseases on the basis of an etiopathogenetic mechanism of immune-mediated damage, such as multiple sclerosis (MS), neuromyelitis optica (NMO), myelin oligodendrocyte glycoprotein antibody disease (MOGAD) [9], neurosarcoidosis, and PACNS [10]. The main feature of inflammatory tumefaction has primarily been large single or multiple lesions in brain parenchyma, atypical for the individual disease under study, sometimes with a size threshold (≥2 cm in demyelinating diseases). Additional features are significant edema and ring-enhancement [9,11]. In some cases, e.g., demyelinating disease, there is a more precise standardization in definition and in neuroimaging, clinical, and biochemical features [12]. 

The definition of t-PACNS is elusive. For example, in a relatively large case series [13], the inclusion criterion for t-PACNS (named “tumor-like mass lesions” by the authors) was “a diagnosis of tumor on brain MRI in the neuroradiological report at disease presentation”. The main MRI feature was the presence of single or multiple contrast-enhanced mass(es) or infiltrative lesion(s), with perilesional edema and mass effect in the absence of ischemic lesions on diffusion-weighted imaging (DWI). An evident limitation of this definition is that it was focused on a single subtype of PACNS, namely, Cerebral Amyloid Angiopathy, Aβ-Related Angiitis (ABRA) [14]. The French cohort [15] was identified according to a similar definition: “brain MRI demonstrating unique or multiple contrast-enhanced mass(es) or infiltrative lesion(s), with perilesional edema and mass effect, not linked to ischemic lesions on DWI”. A more restrictive definition of t-PACNS as tumor-like mass lesion was adopted by Molloy et al. [16], considering only solitary mass lesions. Finally, the case series of Suthiposuwan et al. [10] adopted the inclusion criterion of mass-like brain lesion(s) with a provisional radiologic diagnosis of brain neoplasm on initial MRI and a histopathologic confirmation of t-PACNS and the exclusion criterion of ABRA on histopathology.

One of the mechanisms leading to a mass lesion is a breakdown of the blood–brain barrier of the small vessels via the infiltration of inflammatory cells in the perivascular and parenchymal regions, resulting in a mass-mimicking enhanced lesion [17]. This mechanism could be proposed for tumefactive SV-PACNS, but it is not enough to explain the presence of mass lesions in LV-PACNS. Therefore, several points remain elusive in this pattern of PACNS.

### 2.2. Epidemiology

The rate of t-PACNS has been variably reported in the literature, being proposed as 5% in a mixed case series from two institutions grouped with previously described cases [16]. This paper included only solitary mass lesions, so the prevalence might be underestimated, although being similar in institutional cohorts (7/182 (3.8%) and 1/20 (5%) patients with definite PACNS) and in the literature (30/535 (5.6%) patients). Suthiphosuwan et al. [10] reported multiple lesions in 2/10 cases. The main mimic of a t-PACNS is a tumor, so these patients usually undergo brain biopsy, and this selection bias for biopsy might explain the relatively high finding of SV-PACNS and ABRA in this cohort. Moreover, this bias might justify the definite diagnosis of PACNS [3] among patients with t-PACNS. Molloy et al. [16] reported a median age of 45.5 years (range 6–74) without sex imbalance. Among 45 reported cases of ABRA, 13 (29%) had a presentation with a solitary mass-like lesion and had a median age of 66 years (range 49–74). Another single-institution cohort [13] identified 13/191 (6.8%) patients with t-PACNS, 77% male, with a median age at diagnosis of 66 years (range 25–84 years). Finally, in the French cohort [15], 10/85 (11.8%) patients had a t-PACNS with a median age of 37 years (range 30–48) versus 46 years (range 18–79) in the non t-PACNS patients (*P* = 0.008; 9 [90%]); 60% were males. 

### 2.3. Neuroimaging

The most complete description of neuroimaging features of t-PACNS patients, excluding ABRA and CAA-related inflammation [18,19], was provided by Suthiposuwan et al. [10]. The first suspicion in most patients was primitive intracranial tumors [20]. Suthiposuwan et al. [10] described a case series of 10 patients with the following features: -Lesion number: 8/10 (80%) patients had a single brain lesion and 2/10 (20%) patients had multifocal lesions;-Lesion location: cortex/subcortical white matter in 7/10 patients (70%), deep and/or periventricular white matter in 6/10 patients (60%), basal ganglia in 4/10 patients (40%), and brain stem in 1/10 patients (10%);-Signal features: T1-Weighted Imaging (T1-WI) hypointense signal and intermediate to hyperintense signal on T2WI without diffusion restriction; intralesional microhemorrhages (small linear or punctate patterns on either Gradient Recalled Echo (GRE-T2*) or Susceptibility Weighted Imaging (SWI)) in 8/10 (80%) patients;-Perilesional vasogenic edema (T2-WI high signal): marked in 5/10 (50%) patients and moderate in 5/5 10 (50%) patients;-Enhancement patterns (parenchyma): all patients (10/10, 100%) had enhancing lesions with patchy parenchymal pattern in 5/10 (50%), “mottled appearance” (multiple small hypoenhancing areas within the patchy enhancing masses) in 3/10, small nodular pattern in 5/10, ring enhancement in 2/10, and linear (perivascular) enhancement pattern in 1/10 patients;-Enhancement patterns (leptomeninges): localized leptomeningeal enhancement adjacent to the dominant/largest lesions was found in 4/10 patients (40%), localized subependymal enhancement adjacent to the lesions in 3/10 patients (30%); and diffuse leptomeningeal or subependymal enhancement was not documented;-Vascular imaging (including Computed Tomography Angiography (CTA), Magnetic Resonance Angiography (MRA), and Vessel Wall Imaging (VWI)): CTA or MRA were available in 9/10 patients (90%) of 10 patients with normal findings (notably, none of the patients underwent catheter angiography); VWI was available in 3/10 patients (30%) with normal findings on the imaged proximal large intracranial arteries.

Lee et al. [21] described a series of four patients with tumor-like PACNS considering both baseline and follow-up MRI (21 MRIs for four patients), with a dedicated reading by neuroradiologists assessing the lesions of the brain with respect to their location, number, size, shape, signal intensity, absence or presence of hemorrhage, pattern of enhancement, and changes on the follow-up studies. According to the detailed description of the previous reported cohort [10], the four patients [21] had the following neuroimaging features: -Lesion number: two out of four patients had a single mass lesion and two out of four had multiple lesions (two hemorrhagic lesions in one case and multiple digitate-shaped enhancing lesions with edema in the second case);-Lesion location: single lesions were in the suprasellar region and in the left frontal lobe white matter; multiple lesions were in the right frontal lobe in one case and in the right frontal white matter, corpus callosum, and external capsule in the second case;-Signal features: the single lesions were an enhancing mass with edema in the suprasellar area with decreased signal on apparent diffusion coefficient (ADC) map, and a focal edematous mass-like lesion (2.1 cm) with T2-high T1-low signal lesion, gyral swelling, and increased ADC, respectively;-Enhancement patterns: they were described as variable without further details;-Vascular imaging: one patient had LV-PACNS right M1 middle cerebral artery (MCA) occlusion and left distal internal carotid artery (ICA) stenosis on angiogram; the remaining 3/4 patients had normal vascular imaging.

On DWI and ADC maps, two patients showed a slight decrease in ADC, whereas the other two patients revealed a slight increase in ADC [21].

In the French cohort [16], 10/85 patients were reported as t-PACNS. MRI showed mass lesions in seven out of ten patients (single and multiple lesions in four and three cases, respectively), with edema and mass effect in three and five cases, respectively. In the remaining 3/10 patients, MRI showed white matter infiltrative lesions with edema and mass effect. All patients had a lobar supratentorial location of the lesions. 

In the Mayo Clinic cohort [13], 13/191 patients with mass-like PACNS were reported. This single-center cohort had a wide time range in enrollment and neuroimaging studies were not fully detailed in their features both for the technical parameters and for the findings. In particular, gadolinium-enhancing lesions (intracerebral or meningeal) were reported in 10/13 (76.9%) patients (6/10 intracerebral and 4/10 meningeal enhancement) but it is not known if all patients underwent contrast administration. Two of the enrolled patients had Hodgkin’s lymphoma (in one, HL was simultaneous to PACNS, and in the other, HL was diagnosed 26 years before the PACNS diagnosis). One further limitation is that the reported neuroimaging data were based on MRI at presentation and t-PACNS might develop during the course of PACNS and not necessarily be the presentation pattern. A mass-like lesion was reported on MRI in six out of thirteen patients (a single lesion in three cases and multiple lesions in three other cases), with a conflicting finding of edema and mass effect in five and three cases, respectively. In 7/13 patients, MRI showed white matter-infiltrative lesions with edema and mass effect. This last subset of patients is probably corresponding to the 7/13 patients without parenchymal brain enhancement and it is hard to consider it as t-PACNS according with Suthiphosuwan et al. [10] and Lee et al. [21]. The initial suspicion was gliomatosis cerebri in one patient with a multilobar white matter involvement and a pathological diagnosis of ABRA, and glial tumor in the other ten out of thirteen patients (nine out of ten with a final diagnosis of ABRA), and metastases in two patients, one with infratentorial involvement and one with multifocal supratentorial involvement.

The main features of MRI investigations in t-PACNS series are summarized in Table 2.

In the cohort described by Suthiphosuwan et al. [10], neuroimaging patterns were extensively described, much more than in the other cohorts, from the neuroradiological field. A broad variety of imaging patterns emerged, but few imaging features shared by many of the patients with t-PACNS can be identified and this information is useful for the diagnosis. In particular, solitary lesions are more common than multiple lesions, and the supratentorial cortex/subcortical areas is the most common location [20], with rare involvement of the deep gray nuclei and brainstem. The enhancement of the lesions after contrast administration was constant, and the most common pattern was patchy and nodular enhancement (ring and linear/perivascular enhancement were less common). A subtype of patchy enhancement is represented by multifocal small hypoenhancing areas within the background of patchy enhancement, named “mottled appearance”. There is not a similar description in other case series, and the authors advanced the hypothesis of superimposed small infarctions or necrosis within the parenchymal inflammatory processes as the cause of this appearance. Subependymal or leptomeningeal enhancement, mainly associated with the lesions, was occasionally present. One of the most interesting features was the lack of diffusion restriction in all patients described by Suthiphosuwan et al. [10], but diffusion restriction was reported in other series [10,13,21]. This issue is still open to etiological hypotheses, because the differential effect of SV-PACNS and LV-PACNS on this appearance in t-PACNS is not fully convincing. A more convincing hypothesis could be the venular involvement, a debated issue also in CAA [22], but it needs further investigations, considering the role of perivenular inflammation in other neurological diseases [23]. Another common finding is the presence of punctate or linear microhemorrhages within the lesions on GRE-T2* or SWI sequences [10], probably due to the alteration of vessel wall permeability induced by a vessel wall inflammatory infiltrate.

The findings described by Suthiphosuwan et al. [10] were strongly in support of excluding ABRA/CAA-ri from the t-PACNS patterns. Indeed, the neuroimaging appearance is different, a vasogenic edema without a contrast-enhancing lesion but with predominant leptomeningeal enhancement being the most prominent feature in ABRA/CAA-ri. Therefore, a careful analysis of the neuroimaging pattern supports to consider ABRA/CAA-ri a different subtype of the disease without including it in the t-PACNS pattern. It is possible that the different neuroimaging pattern is due to different pathophysiologic mechanisms within the SV-PACNS subtype. In particular, ABRA and CAA-ri have a similar neuroimaging appearance, but differences in epidemiology and natural history raised the hypothesis that in the first one, inflammation leads to Ab deposition and in the second one, the deposition of Ab in vessel walls because of a decreasing clearance triggers the inflammatory response, with anti Ab autoantibodies being a common finding [18,19,24,25,26]. 

An example of the MRI pattern of t-PACNS is illustrated in Figure 1, Figure 2 and Figure 3. 

### 2.4. Advanced MRI Findings

The first description of the potential use of perfusion and diffusion MRI for PACNS diagnosis is due to Yuh et al. [27] in 1999. In general, a minority of patients described in the literature as having a t-PACNS had advanced MRI available, and, since 1999, there have been very few patients in whom the findings of advanced MRI techniques have been described in detail. In the cohort described by Suthiphosuwan et al. [10], advanced MRI (DSC-MRI) and ^1^HRS-MRI were performed in 3/10 (30%) and 2/10 (20%) patients, respectively. In all MR perfusion studies, low relative cerebral blood volume (rCBV) was found. In Lee et al. [21], one solitary lesion had increased rCBV on perfusion Computed Tomography (CT) in a patient with LV-PACNS, and the case with multiple hemorrhagic mass lesions had decreased CBV on DSC-MRI. 

The low rCBV values could be expected because of the lack of neoangiogenesis in an inflammatory process, unlike malignant brain neoplasms. Even fewer data are available on ^1^HRS-MRI, whose reported pattern was nonspecific and may be less useful than DSC-MRI in the differential diagnosis from a tumor. An example is illustrated in Figure 4. 

Moreover, in one of the cases described by Lee et al. [21], perfusion CT revealed increased Mean Transit Time (MTT) in both MCA territories, but the patient had a LV-PACNS with intracranial steno-occlusions. 

In the largest neuroradiological series [10], with two patients (20%) who underwent MR spectroscopy, the findings were elevated choline peaks compared with the creatine and slightly decreased N Acetyl Aspartate (NAA) peak (choline/creatine ratios were approximately 2.0, and choline/NAA ratios were approximately 0.9); in one patient, a high lipid lactate peak was demonstrated. Spectroscopy found increased choline/NAA and lipid/lactate in the patient with a single lesion and LV-PACNS, and in the patient with multiple hemorrhagic mass lesions [21]. There are a few reports concerning MR spectroscopy features of PACNS. An elevation of the choline/NAA ratio and lipid–lactate peaks in varying-degree, nonspecific findings was reported [10,21] and in a single study [28], a marked elevation of glutamate and glutamine peaks was observed, but not confirmed in other reports [29].

### 2.5. Pathology

Most patients with t-PACNS underwent open brain surgery or biopsy to exclude the main mimic, i.e., a tumor. Then, some information about pathology is available, but the selection bias of patients for biopsy is evident. In the cohort described by Suthiphosuwan et al. [10], 5/10 patients (50%) had a stereotactic-guided biopsy, 4/10 patients (40%) open surgery with gross total resection, and 1/10 patients (10%) an open biopsy. In the entire brain sample, the pathological pattern was a transmural inflammation [30] of small blood vessels in the brain parenchyma and/or leptomeninges, confirming the PACNS diagnosis as definite [3]. The histological pattern was lymphocytic in 9/10 patients (90%). It was characterized by a vasculocentric lymphocytic inflammatory infiltrate, with immunohistochemistry staining positive for CD20 (B-cell lymphocytes) and CD3 (T-cell lymphocytes). In two out of nine patients with lymphocytic vasculitis, the pattern was necrotizing vasculitis. In 1/10 patients (10%), the histopathology was a granulomatous t-PCNSV with vasculocentric predominantly mononuclear cell infiltration, accompanied by multinucleated giant cells and fibroblasts. Due to the exclusion criteria of the case series [10], not considering ABRA, none of these lesions had amyloid depositions on H&E staining. In 7/10, confirmatory immunohistochemistry staining for Ab and/or Congo red were available and normal. In the study by Lee et al. [21], the lymphocytic pattern was documented in all four patients and, even though there were a few granulomas in the perivascular spaces, the granulomas were not an overwhelming feature. In the French cohort [16], all t-PACNS had a biopsy (stereotactic procedure in seven and open-wedge surgery in three), which was positive for PACNS in nine out of ten (eight with a lymphocytic pattern). The negative biopsy contained little brain tissue and the patient had extensive LV-PACNS. Moreover, in four patients, immunohistochemistry studies on biopsy samples combined with gene rearrangement studies excluded lymphoproliferative disease. 

Previous studies [13,16] reported a predominant granulomatous pattern in t-PACNS, but ABRA/CAA-ri cases were included. In fact, in the 13 patients of the Mayo Clinic cohort [13], a positive cerebral biopsy (stereotactic in 10, open-wedge in 3) was reported with a granulomatous pattern in 8/13 patients, accompanied by vascular deposits of β-amyloid peptide in 6/8 patients, a granulomatous and necrotizing pattern in 3/13 patients (1/3 with vascular deposits of β-amyloid peptide), and a lymphocytic pattern in 2/13 patients. Moreover, some of these patients correspond to the patients with ABRA excluded in the definition of Suthiphosuwan et al. [10]. 

## 3. Neuroimaging Clues in the Main Differential Diagnoses

The differential diagnosis of t-PACNS is often challenging, also from a neuroradiological perspective. In fact, the main differential diagnosis of tumor-like mass lesions is a tumor, i.e., Primary Central Nervous System Lymphoma (PCNSL) and glial tumors. Another broad category in the differential diagnosis of t-PACNS is represented by typical and atypical forms of tumefactive demyelination. Neuroimaging is of paramount importance for a correct differential diagnosis and for selecting patients for a brain biopsy. Among neuroimaging techniques, standard and advanced MRI, including functional MRI (DSC-MRI and ^1^HRS-MRI), might provide an insightful contribution. A detailed discussion of each disease is outside the scope of this review. For this reason, some diagnostic macrocategories will be addressed rather than individual diseases.

Neuroimaging techniques are fundamental for the differential diagnosis of tumor-like brain lesions. In particular, the combination of conventional and advanced MRI has been widely used, in particular in the study of primary CNS tumors. In other non-neoplastic pseudotumor diseases, advanced neuroimaging techniques have been used occasionally, so there are fewer data available. For this reason, it seems appropriate to briefly detail the advanced MRI techniques used for tumor diagnosis, the application of which will be proposed in non-neoplastic pathologies to support differential diagnosis. In addition, standard MRI sequences, such as DWI/ADC, might provide useful insight into the differential diagnosis of tumor-like brain lesions, and their relevance deserves a mention together with advanced MRI techniques. Most of these techniques have been addressed by global initiatives aiming to pursue the standardized development of physiological imaging biomarkers. Two of the most relevant initiatives are the Quantitative Biomarkers Alliance (QUIBA) and the American College of Radiology Imaging Network (ACRIN). 

### 3.1. DWI/ADC and Advanced MRI Techniques 

DWI uses the Brownian motion of water within tissue to derive microstructural information. The dominant signal comes from the interstitial compartment, including the extracellular (ECS) and extravascular space (EVS). ADC is the quantification of this information [31]. Several tissue components may affect the diffusion image signal, in particular cell density and matrix composition. Indeed, high cellular lesions (e.g., tumors) are characterized by low diffusivity. Standard brain tumor imaging includes three-directional DWI performed with two or three b values, typically b0, (+/−b500) and b1000 mm/s^2^. The mathematical subtraction of T2 effects from DWI acquisition allows the calculation of the ADC map, providing quantitative diffusion measurement to aid in the differential diagnosis of tumor-like lesions. Low diffusivity might be expressed by malignant tumors and also by benign and non-tumoral lesions. As for the other MRI sequences, the DWI/ADC data should be interpreted together with morphological sequences. Moreover, DWI is the best modality to identify cytotoxic edema and distinguish infective lesions from a brain tumor.

Increasing the number of examined diffusion directions to six or more is the basis of diffusion tensor imaging (DTI). DTI provides information about the directionality of water diffusion, moving water within tissues preferentially into some directions (e.g., along axons). This phenomenon is called fractional anisotropy (FA) and its quantification allows to reconstruct DTI data and to visualize peritumoral white matter tracts. Among DWI techniques, there are also some more complex ones, based on the assumption that water diffusion has a non-Gaussian distribution, including compartmental diffusion modeling and kurtosis imaging, which will not be considered for this review.

The grade of vascularization is another biological feature potentially explorable using perfusion MRI techniques in brain lesions (tumors are usually hypervascular lesions). In fact, in high-grade gliomas, the increase of rCBV and relative cerebral blood flow (rCBF) is due to neovascularity, but this feature has limited specificity for diagnosis in isolation. Then, perfusion imaging should be interpreted in conjunction with structural imaging. The three main MRI perfusion methods are as follows: Dynamic susceptibility contrast (DSC) imaging, which is currently the most commonly used technique;Dynamic contrast-enhanced (DCE) imaging (both requiring intravenous administration of a gadolinium-based contrast medium);Arterial spin labeling (ASL), not requiring contrast media.

DCE- and DSC-MRI will be discussed in the differential diagnosis of tumor-like brain lesions. 

The DSC-MRI technique uses a transient vascular signal reduction on T2*-WI during the first pass of a gadolinium bolus injection. Indeed, a volume of T2* images is repeatedly acquired at rapid (1–2 s) intervals before and during the contrast injection for 1–2 min in total. For tumor imaging, the rCBV is the most used and best validated DSC parameter. Several factors may affect perfusion measurement in acquisition and postprocessing (e.g., the presence of marked susceptibility artifacts from intratumoral hemorrhage). Therefore, a standardized imaging protocol was proposed in 2015, which may include a preload bolus to minimize T1 leakage effects in the presence of a disrupted blood–brain barrier [32]. 

The DCE-MRI technique measures the increase of signal intensity on T1-WI from baseline after gadolinium administration over a certain time period (typically around 5–7 min). The resulting time–signal intensity curve depends on several factors (e.g., tissue perfusion, vascular permeability, the concentration of gadolinium in the intravascular and extravascular-extracellular space, etc.). The shape of the time–signal intensity curve may be used to identify a rapidly enhancing hyperperfused tumor or a slow leakage of contrast medium into the extravascular extracellular space. Quantitative measures are available through mathematical modeling (e.g., the Tofts model and extended Tofts model). Among these, the transfer coefficient K_trans_ depends on endothelial permeability, vascular surface area, and blood flow within the tissue of interest. Plasma volume (V_p_) correlates with DSC-derived rCBV, and the volume of the extravascular extracellular space (V_e_) correlates with tumor cellularity. DCE is not compromised by intralesional susceptibility artifacts, which can be significant following surgery, and has the advantage of increased spatial resolution. 

Finally, ASL uses magnetically labeled blood as an endogenous tracer to assess tissue blood flow. This technique requires no contrast injection and is unaffected by intra- or perilesional susceptibility effects, being suitable for the pediatric population and patients in whom gadolinium injections should be avoided. 

Functional information about brain lesions might also come from examining the tissue metabolite distribution using spectroscopy. This technique can be performed as a single- or multivoxel analysis. The first one is technically easier to achieve but assumes the average values of metabolites within the measured area. Multivoxel spectroscopy is able to identify spatial variations, but good quality spectra are more complex to achieve. In neuro-oncology, MR spectroscopy (^1^HSR-MRI) has a recognized role in the distinction of a tumor from non-neoplastic conditions. On spectroscopy, the signature of tumors is an elevation of choline (Cho) and a decrease in N-acetyl aspartate (NAA), a neuron-specific marker, and creatine (Cr). Depending on tumor type, lactate or lipids can be present. The peak of Cho is due to the accelerated synthesis and destruction of cell membranes in growing tissues, which is amplified in areas of high mitotic activity. Conversely, lactate accumulation is a result of nonoxidative glycolysis (Warburg effect), which produces a tissue milieu prone to angiogenesis. In addition, ^1^HSR-MRI spectra are influenced by the choice of echo time (TE): using a short TE (20–40 ms), smaller metabolites (myo-inositol, glutamate/glutamine, and lipids) are better demonstrated but with a less stable baseline; an intermediate (135–144 ms) TE allows for a less distorted baseline with improved NAA and Cho quantification with lactate appearing (inconsistently) inverted from the baseline; long (270–288 ms) echo times tend to be less widely used for brain tumor imaging.

Among the advanced MRI techniques, an increasingly relevant role is played by the search for intratumoral susceptibility signs (ITSSs) with SWI sequences. SWI is a 3D high-resolution gradient echo sequence that uses both phase and magnitude images, and contains a reconstruction of the minimum Intensity Projection (minIP) and MultiPlanar Reconstruction (MPR) techniques with 3–10 mm images, with high sensitivity to the blood, iron, and calcification in the tissue which builds-up susceptibility [33]. Signal intensity in SWI is influenced by hematocrit, deoxyhemoglobin concentration, erythrocyte integration, clot structure, molecular diffusion, pH, heat, voxel size, contrast material, blood flow, and vessel orientation [34]. Moreover, the phase component of the SWI sequence can be used for recognizing calcification, which cannot be separated from the hemorrhage using GRE [35]. High-grade gliomas are characterized by a relative increase in deoxyhemoglobin concentrations due to angiogenesis and an increased vascularization of the tumor, leading to a signal loss due to a susceptibility effect in SWI [36]. Moreover, the presence of ITSS might be considered as an indicator of tumor grading while the absence of ITSS was related to non-tumoral lesions [36,37,38]. The ITSS scoring system is a semiquantitative scale: grade 0 signifies no ITSS; grade 1, 1–5 ITSSs; grade 2, 6–10 ITSSs; and grade 3, 11 or more ITSSs [38].

### 3.2. Primary Central Nervous System Lymphoma 

PCNSL is probably the main differential diagnosis of t-PACNS, being restricted to the brain, spinal cord, or eyes [39]. PCNSL accounts for 1–5% of all brain tumors [40], and it might involve both immunocompetent and immunocompromised patients. In particular, the incidence rate of PCNSL is increasing in immunocompetent patients [40]. Neuroimaging is crucial for diagnosing PCNSL [41], but none of the CT or MRI findings might unequivocally differentiate PCNSL from other brain lesions. The appearance of PCNSL on neuroimaging is mainly due to its hypercellularity, high nuclear/cytoplasmic ratio, the disruption of the blood–brain barrier, and its predilection for the periventricular and superficial regions, often in contact with ventricular or meningeal surfaces [42].

The classical MRI appearance of PCNSL in non-AIDS patients is as a solitary homogeneously enhancing parenchymal mass, but multiple lesions are reported in 20–40% of cases and a ring-like enhancement in 0–13% [43]. Moreover, linear enhancement along perivascular spaces is highly suggestive of PCNSL [44]. Perifocal edema is usually present but less prominent than that in malignant gliomas or metastases. The most frequent locations are the central hemispheric or periventricular cerebral white matter and a superficial location adjacent to the meninges. Frontal lobe, basal ganglia, brain stem or cerebellum, and spinal cord are involved in 20–43%, 13–20%, 9–13%, and 1–2% of PCNSLs, respectively [40]. An extension of parenchymal PCNSL to the eye has been reported in 25% of the PCNSLs [45], but primary intraocular lymphoma is a very rare subset of PCNSL. If present, a vitreal localization allows the diagnosis by means of a cytologic examination of vitreal aspirate. However, MRI has a low sensitivity for intraocular lymphoma, and negative findings do not preclude the intraocular involvement of PCNSL. Primary dural lymphoma is a rare subtype of PCNSL arising from the dura mater and presenting with a CT or MRI appearance of diffusely enhancing single or multiple extra-axial masses, potentially mimicking meningiomas. In this subtype of PCNSL, the most involved site are the cerebral convexities, but the falx, tentorium, sellar/parasellar regions, or spine might also be involved.

In immunodeficient patients, PCNSL is most frequent and often multifocal at diagnosis. Indeed, multifocal lesions have been reported in 30–80% of patients with AIDS-related PCNSL [46]. The presence of necrotic regions in many lesions accounts for irregular, peripheral, or ring-like contrast enhancement. The basal ganglia and corpus callosum are frequently involved, and spontaneous intralesional hemorrhage is more frequent in AIDS patients than in non-AIDS patients.

A neuroradiological review [47] of the CNS lymphomas included an update of the WHO’s classification of brain tumors 2021 [48] and hematolymphoid tumors 2022 [49]. The authors were against the use of the PCNSL term, because its description refers to the most common presentations of the most frequent type, i.e., primary diffuse large B-cell lymphoma (DLBCL) of the CNS, negative for Epstein–Barr virus (EBV), and not to all CNS lymphomas. The same lymphoma might have atypical patterns, and other specific subtypes have different imaging features. According to the updated classification [48,49], the recommended nomenclature is not PCNSL, but primary DLBCL of the CNS. However, the literature from the period before 2021 refers to the PCNSL, so in this description, we have also maintained the reference to the old nomenclature. Primary DLBCL of the CNS accounts for 80%–85% of all CNS lymphomas, occurs almost always in immunocompetent patients, is EBV-negative, and is of unknown etiology. The neuroradiological appearance is usually as single or multiple (30–35%) supratentorial (>80%) parenchymal lesions, with a particular affinity for the basal ganglia, periventricular regions, midline, and corpus callosum (about 45%). The posterior fossa is rarely involved, and a spinal location is exceptional. Associated leptomeningeal or subependymal enhancement is characteristic, but not exclusive. The typical perivascular pattern on histopathology is the basis for the characteristic perivascular enhancement on imaging. MRI features of DLBCL are hypointensity on T2WI with marked hyperintensity on DWI. The main alert on MRI interpretation is that, as is well known [50], a T2-blackout effect (persistent hypointensity on b = 1000 images due to very low T2 signal) may occur, and then the assessment of diffusion restriction is more reliable on ADC map hypointensity than on b = 1000 hyperintensity [41,47,51]. Notably, a low T2 signal and diffusion restriction (in addition to non-contrast CT (NCCT)_hyperattenuation) correlate with high cellularity on histology and high proliferation indexes [52].

Another relevant issue is the presence of hemorrhage or necrosis on preoperative studies, in particular in immunocompetent patients, confirmed by the histologic appearance of tumor samples. MRI identifies hemorrhage in up to 50% of patients evaluated with SWI (20% with T2WI) and heterogeneous or ring-like contrast enhancement in up to 10–15% of cases [53,54]. The ring-like contrast enhancement is usually associated with necrosis. Actually, the presence of signs of intralesional hemorrhage or necrosis is no longer considered an alert for alternative diagnoses. 

The main features of PCNSL on conventional MRI are proposed in Figure 5.

Standard MRI is not able to differentiate PCNSL from other neoplasms and non-neoplastic diseases, including t-PACNS, because the typical imaging characteristics may not be present [40]. A relevant contribution for the differential diagnosis of tumefactive brain lesions is provided by DWI, DSC-MRI, and MR spectroscopy. DWI might provide a surrogate measure of cellularity, so, as PCNSL is highly cellular, it is usually hyperintense on DWI and hypointense on ADC maps [55,56]. This appearance is not unique for PCNSL, but it is also present in acute ischemic stroke, brain abscesses (central necrotic core), high-grade gliomas (solid portion), and some metastases [56]. Usually, PCNSL lesions have a more restricted diffusion and lower ADC values than high-grade gliomas (HGGs) and metastases [57]. Low ADC values are predictive of shorter progression-free survival and overall survival. The time course of ADC measurements may be used in the surveillance of therapeutic response [58], assuming an inverse correlation between ADC values and the cellular density of the tumor.

Quantitative advanced MRI techniques beyond DWI, ^1^HSR-MRI and DSC-MRI are included in consensus recommendations for imaging CNS lymphoma [59], showing promising results for presurgical diagnosis. In the above cited consensus recommendations, a careful and precise technical framework was created to standardize MRI investigation at 1.5 T and 3.0 T, paying attention to pulse-sequence parameters (TE, TR, flip angle), prebolus usage, and leakage corrections for DSC-MRI. In recent years, DSC-MRI gained attention and evidence of usefulness in neuro-oncology, measuring rCBV as a surrogate noninvasive imaging biomarker of tissue microvascular volume [60,61]. Indeed, unlike glioblastoma, PCNSL neoangiogenesis is minimal and often lacks a neurovascular unit. The main feature of PCNSL is low-to-intermediate CBV, a high percentage of signal recovery (PSR), and characteristic time–intensity curve morphology [51,62]. In addition, the maximum rCBV measured in tumor tissue is lower in lymphomas than in other brain tumors, helping in differentiating PCNSL from glioblastomas and metastases [63]. Moreover, it has a characteristic time–intensity curve (TIC), due to a massive leakage of contrast media into the interstitial space [64], whose evaluation is performed through normalization (nTICs) [51,65]. Further studies are needed to assess the role of DSC-MRI in the characterization of disease burden, molecular stratification, and contribution to response assessment. In particular, the assessment of rCBV has been demonstrated to be useful in the pre-operative differentiation of PCNSL from glioblastoma, despite the variability of PCNSL rCBV values.

DCE-MRI might provide some measure of microvascular permeability in tumor tissue [66]. Among all DCE-MRI parameters, K_trans_ has most consistently demonstrated its value in distinguishing differential diagnoses. The finding of higher values of K_trans_ in lymphoma than in other brain lesions (glioblastoma and metastases) has been confirmed in several independent studies, and is consistent with the findings of DSC-MRI [67,68]. In addition to K_trans_, the extracellular volume fraction or the volume of the extravascular extracellular space (V_e_), correlating with tumor cellularity, is consistently higher in lymphoma than in glioblastoma and brain metastases, and may differentiate lymphomas from other brain tumors [69]. However, while DSC represents the most commonly employed technique to assess tissue/tumoral perfusion in the brain, it remains an optional sequence for patients with PCNSL [59]. For data analysis, Quantitative Imaging Biomarkers Alliance^®^ (QIBA) recommends the use of a DCE-MRI tool that computes the pharmacokinetic parameters (K_trans_ and V_e_) using the standard Tofts model [66].

In PCNSL, ^1^HSR MR spectroscopy has demonstrated elevated lipid peaks combined with high Cho/Cr ratios, as in glioblastoma and metastases [55]. Spectroscopy can also support the diagnosis with two issues: short TE found much lower mIns (described as a glial marker) than that associated with enhancing non-necrotic astrocytoma (ie, grade 3), and long TE found much lower mobile lipids (associated with necrosis) than glioblastoma or metastasis [70]. DTI is a sensitive tool for the detection of changes in white matter structure [57]. A quantitative FA map shows hypointensity corresponding to decreased FA values in most brain tumors, but the FA values of PCNSL are significantly lower than those of glioblastoma.

The main findings on advanced MRI in PCNSL are illustrated in Figure 6. 

Addressing all subtypes of lymphomas with CNS involvement is beyond the scope of this review. Mention will therefore be made only to some diseases, much rarer than PCNSL, which can enter into differential diagnosis with t-PACNS. 

According to the latest WHO classification [59], the immunodeficiency-associated CNS lymphoma subtype, specifically corresponding to primary DLBCL of the CNS, is EBV-positive. It represents 8–10% of all primary CNS lymphomas and it is the second most frequent type of primary CNS lymphoma. Its clinical context has changed during recent decades, with a shift from HIV-related immunodeficiency to post-transplant status, autoimmune disease, and iatrogenesis [71,72,73]. In particular, immunosenescence [74] has been invoked as a potential cause of immunodeficiency-associated CNS lymphoma and, considering the wide age range of PACNS (and t-PACNS), sometimes this subtype of lymphoma enters in differential diagnosis, together with glioblastoma or metastases, more frequently than opportunistic infections. The location can be deep or hemispheric, usually with multiple lesions. On MRI, the main features are a large necrotic core, ring-loke contrast enhancement, and intermediate-to-prominent signs of hemorrhage. The signal pattern on T2-WI and DWI is variable, making this tumor more similar to glioblastoma and metastasis. The necrotic core has a T2-WI heterogeneous hypointensity not corresponding to blood products or mineralization in the non-enhancing portion, whereas other tumors have hyperintense T2 signal of non-hemorrhagic necrosis [75].

In this regard, advanced MRI techniques, such as DSC-MRI, might provide some useful information in differential diagnosis [75]. The perfusion features are of low-to-intermediate CBV, high PSR, and a characteristic TIC morphology in the solid parts of the tumors. The authors of [75] found significant differences between DLBC EBV-positive CNS lymphoma and glioblastoma or metastasis at almost all time points of the TIC, with the greatest level around the maximal-signal-intensity drop and signal-recovery segments, reinforcing the visual assessment. The ^1^HSR-MRI pattern is conditioned by prominent mobile lipids overlapping with necrotic glioblastomas or metastasis [70].

A relevant issue is that the neuroimaging investigation of lymphomas involving the CNS is well standardized, evidence-based [59] and homogeneous, allowing a reliable comparison between individual centers. Unfortunately, this is not the case with PACNS. 

Finally, PCNSL lesions may not contrast-enhance and can show atypical features of hemorrhage, calcification, cyst, and necrosis, mimicking non-malignant CNS diseases. Several studies suggested the usefulness of DWI-derived ADC in differentiating lymphoma from other diseases, including glioblastoma [76] and TDLs [77]. When compared to these CNS conditions, PCNSL lesions show lower ADC values, thereby reflecting their higher cellularity. In the case series of Lin et al. [78], aiming to assess the role of ADC quantification for this purpose, the ROIs were placed avoiding cystic, necrotic, and hemorrhagic regions in enhancing lesions and, for non-enhancing lesions, in the areas showing restricted diffusion. Interestingly, only 3 (18.8%) patients were diagnosed with lymphoma on neuroimaging, and in the remaining 13 cases, 9 were misdiagnosed as glioblastoma; the remaining patients were misdiagnosed as demyelinating disease (n = 2), vasculitis (n = 1), and meningioma (n = 1). Atypical neuroimaging features were present in about a quarter of the patients, and histology review suggested that the majority of these lesions contained hemorrhage (eight out of nine with GRE sequences) and necrosis (four out of six with central non-enhancement). Despite a lack of restricted diffusion on gross examination of DWI in some, all lesions had low ADC values, and this supports the utility of quantitative ADC measurement in the diagnosis of PCNSL. The authors found median ADC_min_ values of 0.70 × 10^–3^ mm^2^/s, very similar to the cutoff value of 0.69 × 10^–3^ mm^2^/s proposed by Ahn et al. [79] as able to distinguish lymphoma from GBM with 87% sensitivity and 88% specificity. In distinguishing PCNSL from glioblastoma, the combination of DSC-MRI and DCE-MRI parameters showed a better performance than single-parameter rCBV analysis [80].

### 3.3. Tumefactive Demyelination Lesions

Tumefactive demyelinating lesions (TDLs) are roughly intended as large demyelinating lesions with significant mass effect and surrounding edema. They are seen in multiple sclerosis (MS), but also in other conditions, including neuromyelitis optica spectrum disorder (NMOSD), Baló concentric sclerosis (BCS), myelinoclastic diffuse sclerosis (Schilder disease), acute disseminated encephalomyelitis (ADEM), acute hemorrhagic leukoencephalitis, and autoimmune-mediated encephalitis. TDLs have been arbitrarily defined according to a major diameter greater than 2 cm on T2-weighted MRI [81], although smaller lesions between 0.5 and 2 cm may also show similar MRI characteristics and clinical evolution [82]. They might be included in the differential diagnosis of brain tumefactive lesions, such as t-PACNS. As for PCNSLs, a detailed description of the pathophysiology and neuroimaging appearance of each disease associated with TDLs is outside the purpose of this review, so all diseases will be included in the neuroradiological description [83] of TDLs.

The real prevalence of TDLs is unknown. Some recent cohort of MS patients reported a prevalence ranging from 1.4% (10/711 patients) to 8.2% (24/293 patients) [84,85] and a mean age between 20 and 40 years, with female predominance [11,86]. 

Conventional MRI and advanced MRI techniques show typical findings in TDLs and might be useful for a differential diagnosis [87,88,89,90]. The main features of TDLs on conventional MRI have variable frequencies [81], as will be detailed in the following paragraph. TDLs show different patterns of contrast enhancement, including homogeneous, heterogeneous, patchy and diffuse, cotton-ball, closed-ring, open-ring, and nodular [11]. Among these patterns, the most frequently reported one is an open ring of enhancement with the incomplete portion of the ring close to the gray matter of the cortex or basal ganglia [91]. According to the main hypothesis, the enhancing component of the ring could be an advancing front of demyelination, whereas the non-enhancing central core might represent a more chronic inflammatory process [92]. The histological correlate of an open-ring enhancement is infiltration by macrophages and angiogenesis at the inflammatory border [93]. Most TDLs are supratentorial, with frontal lobe (incidence 40.7–56%) and parietal lobe (incidence 42–74.1%) being the most common sites [11,94], followed by corpus callosum, occipital, and temporal lobe. As for PCNSL, the hypoattenuating appearance on NCCT of the areas of enhancement on MRI may be useful to better distinguish TDLs from tumors. Therefore, CT plus MRI showed stronger diagnostic accuracy than MRI alone (97% vs. 73.0%), with a sensitivity of 87% and a specificity of 100% [95].

As previously said, conventional MRI findings (open-ring or incomplete-rim enhancement, a T2 hypointense rim, absent or mild mass effect, and absent or mild perilesional edema) demonstrate variable frequencies [94]. TDLs can be classified into four different subtypes based on the most prominent conventional MRI characteristics [96]: megacystic, Balò-like (lesions with multiple concentric/alternating bands of signal intensity), infiltrative (large, ill-defined areas of T2 abnormalities), and ring-like (round lesions with ring-like enhancement). However, 14–17% of cases may not fit any of the categories [96,97]. 

Several studies reported the use of advanced MRI techniques, including DWI, DSC-MRI, ^1^HRS-MRI, and DTI. The diagnostic performance of these techniques for differentiating TDLs from primary brain tumors have been systematically reviewed [90]. The conventional MR imaging findings of TDLs are as follows: -An open-ring or incomplete-rim enhancement;-A closed ring or complete rim enhancement;-A T2 hypointense rim;-An absent or mild mass effect;-Absent or mild perilesional edema.

Their pooled incidence is reported in Table 3. 

The global diagnostic performance of MRI in differential diagnosis between TDLs and primary brain tumors was addressed in a metanalysis [90] and reported as a pooled sensitivity of 89% (95% CI, 82–93%) and a pooled specificity of 94% (95% CI, 89–97%). The study confirmed that, in routine clinical practice, the differentiation of a TDL from a primary brain tumor is challenging and some MRI findings might be overestimated in their incidence. Open-ring or incomplete-rim enhancement was proposed as highly specific (94%) in a small case series [91], but the incidence of open-ring or incomplete-rim enhancement was 35% (95% CI, 24–47%), although it was significantly higher than the incidence of closed-ring or complete-rim enhancement (18% [95% CI, 11–29%]). Then, open-ring or incomplete-rim enhancement may be useful for differentiating a TDL from a primary brain tumor, when present, but no information is available for other tumefactive diseases such as t-PACNS. In some papers, the mass effect of perilesional edema has been graded, and classified as mild (mild sulcal effacement), moderate (ventricular impingement), or severe (subfalcial herniation) [98]. In the series by Ayrignac et al. [99], the lesions, involving mainly white matter, were frequently juxtacortical (90%) and periventricular (80%). The extension to the cortex and deep grey matter was relatively infrequent (35 and 20%, respectively). Corpus callosum involvement was observed in 12 patients (60%), with a butterfly-like pattern in 8 (40%). A total of 80% of the lesions had associated edema, with mild or moderate mass effect in 87.5% of cases and severe mass effect in 12.5%. A lesional peripheral rim was observed as T2 hypointensity in 15% of cases and as DWI hyperintensity in 70% [99]. The central core was hyperintense and iso/hypointense on DWI in, respectively, 50% and 45% of cases. The corresponding ADC map of the central core was hypointense in 20% of patients, with a peripheral hypointense rim in 40%. Notably, 20% of patients had T2* hypointensities suggestive of hemorrhage. All lesions had gadolinium enhancement, peripheral in 18/20 patients (90%), with open ring in 15 patients (75%), and a central, heterogeneous enhancement in 63%.

The studies using advanced MRI techniques for differentiating TDLs from PCNSLs are few and proposed significantly higher ADC_min_ in TDL than in PCNSL [77,98], with one study demonstrating that ADC_min_ with a threshold of 0.556 × 10^−3^ mm^2^/s was the best indicator for differentiating TDL from atypical PCNSL (with a sensitivity of 81% and a specificity of 89%) [77]. Another study showed a lower choline/NAA ratio in TDLs than in PCNSLs, with a threshold for differentiation of 1.73 (with a sensitivity of 89% and a specificity of 76%) [77]. Another finding is the NCCT hypoattenuation of MRI-enhanced regions, observed in 93% of TDL cases, but only 4% of primary brain tumors [97]. 

The use of ADC values has been investigated to distinguish TDLs from primary brain tumors. TDLs usually show heterogeneous ADC values, higher in the center of the lesion due to vasogenic edema and myelin destruction, and lower peripherally due to inflammatory infiltrates [100]. Another feature is the dynamic evolution of restricted diffusion on serial MRI scans [101]. Moreover, TDLs may show a reduced isotropic component of the diffusion-tensor (p) within the enhancing part of the lesion, and the mean value of anisotropic component of diffusion-tensor (q) is reduced in tumefactive demyelination compared with gliomas [102]. However, the minimum ADC values are usually higher in TDLs than in lymphomas [88,103]. Finally, compared with high-grade gliomas, the peripheral enhancing portion of TDLs shows a higher fractional anisotropy but lower mean diffusivity values [89].

DSC-MRI has been rarely used to compare the rCBV of TDLs with the rCBV of intracranial neoplasms in a small subset of patients and with variable results. Indeed, TDLs might have a lower or similarly increased rCBV compared with intracranial neoplasms [104].

On ^1^HRS MRI, TDLs may show an increased Cho-containing peak and a decreased N-acetyl aspartate (NAA) peak (i.e., an increased Cho/NAA ratio). This pattern is similar to the one found in tumors, but two small cohort studies found an almost identical cutoff of the Cho/NAA ratio > 1.72 or >1.73 as an indicator of high-grade gliomas rather than TDLs [77,105]. However, the practical applicability of this cutoff is questionable. When used as an adjunct to conventional MRI, the Cho/NAA ratio has been shown to improve diagnosis [77]. Some studies have shown that a lactate peak and glutamate/glutamine peak may also be present in TDLs [106].

Sometimes, TDLs might have an atypical appearance, in particular with infiltrative lesions involving the corpus callosum, challenging the differential diagnosis towards a brain tumor. In these cases, brain biopsy is frequently needed [80,107,108]. 

An example of TDL appearance on MRI is illustrated in Figure 7. 

### 3.4. High-Grade Gliomas

In Europe, approximately 80,000 new cases of primary brain tumors are expected each year, with a predicted increase due to an aging population [109]. In adults, the most common malignant primary brain tumors are gliomas, with an average annual age-adjusted incidence rate of 5 per 100,000 population [110,111]. High-grade gliomas (HGGs) or glioblastomas are the focus of differential diagnosis with tumor-like brain lesions. They are classified within the adult-type diffuse gliomas in the recent update of the WHO classification of CNS tumors [112] and are the most commonly occurring malignant primary CNS tumors, accounting for 49.1% of malignant primary tumors and 14.3% of all primary CNS tumors [110]. The peak age-adjusted incidence is 3.23 per 100,000 persons per year. Glioblastomas are 1.6 times more common in males than in females, and the median age at diagnosis is 65 years. For the purposes of the differential diagnosis of tumor-like brain lesions, only WHO grade III/IV gliomas or high-grade gliomas (HGGs) are considered, and in particular glioblastomas. 

Conventional MRI is the standard imaging method in neuro-oncology. Current best practice recommendations for a brain tumor MRI protocol are to include volumetric (3D) T1-WI before and after Gadolinium contrast injection, 2D or 3D axial FLAIR, 2D axial DWI, and axial T2-WI using a 1.5 or 3T MR system. Frequently, these tumors are located in the subcortical white matter and deeper gray matter of the cerebral hemispheres, although they can infiltrate the adjacent cortex and the corpus callosum into the contralateral hemisphere. Glioblastomas can also affect the brainstem, cerebellum, and spinal cord. DWI may be helpful to differentiate glioblastomas from PCNSL or cerebral abscesses, both of which may demonstrate restricted central diffusion [113]. According to the 2021 WHO classification [112], a HGG that meets the molecular definition (TERT promoter variation, EGFR gene amplification, and +7/−10 chromosome copy-number changes) but lacks the classical histologic microvascular proliferation and necrosis is often referred to as a molecular glioblastoma and may show only faint or patchy contrast enhancement or may lack central necrosis. Glioblastomas are often unifocal, but there can be smaller satellite areas of enhancement. Occasionally, MRI can show a non-enhancing or minimally enhancing lesion in an adult patient with a newly onset seizure, and the lesion can rapidly evolve in weeks to a ring-enhancing, necrotic lesion. 

ADC values have been shown to correlate inversely with tumor cellularity because high cellularity reduces the EVS and limits the diffusion of water within it [114,115]. The ECS of HGG is characterized by an overproduction of extracellular matrix (ECM) glycoproteins, e.g., tenascin, preventing water diffusion [116]. While cellular density is generally inversely related to ADC, it should be noted that diffusivity is also markedly dependent on the composition of the extracellular matrix [117]. Irrespective of detailed histological grade, there is consistent evidence in the literature of higher ADC values for LGG when compared with HGG [118,119]. Lowest ADC values were associated with IDH-WT glioma [118,120,121,122], and IDH-mutant gliomas with a 1p/19q co-deletion had significantly lower ADC values (and thus higher cellularity) than the 1p/19q intact group [123]. The majority of published data on ADC in glioma were derived from the isotropic monoexponential analysis of b = 0, 1000 acquisitions, but advanced quantitative diffusion imaging studies are also available with non-monoexponential models and DTI [124].

DSC-based rCBV and endothelial permeability are useful in discriminating WHO grade I, II, and III gliomas, proposing rCBV as more indicative of a HGG than endothelial permeability [125]. Although these findings suggest that higher rCBV values are associated with less aggressive histological and genetic (oligodendroglioma; 1p/19q co-deleted) subtypes, high rCBV values are also associated with HGGs [125]. Summarizing, elevated CBV in a tumor of unknown histological subtype could indicate 1p/19q co-deleted oligodendroglioma (IDH-mutated) that carries a good prognosis, an IDH mutant non-co-deleted astrocytoma with anaplastic elements carrying a poor prognosis, or an IDH-WT glioma with a very poor prognosis. The main information is that CBV is a limited diagnostic or predictive tool when considered in isolation. However, DCE-derived parameters might be used to stratify glioma grades with high accuracy [125]. At the moment, DSC-derived measures of CBV have been more extensively validated than DCE- and ASL-derived parameters in the characterization of gliomas [126]. 

Spectroscopy is frequently used in the investigation of tumors and tumor-like lesions, with the baseline hypothesis that differences in the spectra derived from normal brain tissue and a brain tumor reflect tumor-related metabolic processes, involving NAA and Cho, Cr, lactate, myo-inositol, and lipid moieties [31,127]. In two studies [128,129], metabolite ratios Cho/Cr and Cho/NAA were individually capable of distinguishing HGGs from LGGs better than conventional imaging, but the diagnostic performance of DSC-derived rCBV measures was superior. A recent meta-analysis by Wang Q. et al. included studies from single- and multivoxel MRS at 1.5 and 3 T to assess the diagnostic accuracy of differentiating HGGs from LGGs, reporting a pooled sensitivity of 80% and a specificity of 76% for Cho/NAA, 75% and 60% for tCho/tCr, and 71% and 70% for NAA/tCr [130].

An example of conventional and advanced MRI in HGG is provided in Figure 8. 

### 3.5. Neurosarcoidosis

Tumefactive brain lesions are a rare manifestation of neurosarcoidosis, but they represent a challenging scenario because of the overlapping features with primary and metastatic CNS tumors and sometimes with t-PACNS [131]. In fact, leptomeningitis is the main neuroimaging manifestation of neurosarcoidosis, being present in up to two thirds of patients and potentially in the differential diagnosis with the leptomeningeal spreading of systemic cancer [8]. Enhancing brain parenchymal lesions occur in 10–27% of general neurosarcoidosis cohorts, and intraparenchymal tumefactive lesions are even rarer (6–8%) and usually less than 1 cm in size [132]. The retrospective series proposed by Bou et al. [8] aimed to better define tumefactive neurosarcoidosis. The MRIs of patients with a diagnosis of neurosarcoidosis were reviewed for the following inclusion criteria: (1) the lesions were within brain parenchyma, (2) the lesion’s largest dimension was greater than 1 cm, and (3) perilesional edema and/or mass effect were present. The morphological classification of the lesions separated them into spherical (i.e., circular or oval in appearance) and stellate (i.e., star-like with arms radiating from a central axis), according to the 2018 Consensus Diagnostic Criteria from the Neurosarcoidosis Consortium [133]. Only 9/214 patients (4.2%) met the criteria for inclusion in the study; they had a median age of 37 years (range 23–47 years, standard deviation 8.3 years) at the onset of clinical manifestations of neurosarcoidosis and were predominantly African American (8/9, 88.9%) and female (5/9, 55.6%). All cases were biopsy-confirmed, and in eight out of nine patients (88.9%), brain parenchymal tumefactive lesions, the initial presenting feature of sarcoidosis, were present. Concerning neuroradiological features, multiple tumefactive lesions were present in four out of nine (44.4%) patients, lesions were spherical in seven out of nine patients (77.8%) and stellate (and single) in the remainder. Perilesional edema was present in all cases, which was extensive in five out of nine (55.6%). Five cases (5/9, 55.6%) demonstrated mass effect, including four affecting the ventricles and one resulting in uncal and subfalcine herniation. Contrast enhancement patterns were heterogeneous in five out of nine (55.6%), with well-demarcated borders in six out of nine (66.7%). No lesion exhibited diffusion restriction or hemosiderin deposition. Lesions were most common in the supratentorial brain (11/16, 68.8%): frontal lobe (5/16, 31.3%), basal ganglia (2/16, 12.5%), cerebellum (2/16, 12.5%), pons (2/16, 12.5%), subinsular region (2/16, 12.5%), midbrain (1/16, 6.3%), temporal lobe (1/16, 6.3%), and occipital lobe (1/16, 6.3%). Associated cranial radiographic features included leptomeningeal enhancement in seven out of nine (77.8%) and pachymeningeal enhancement in two out of nine (22.2%). Spinal MRIs were obtained in four patients, and a thoracic spinal cord parenchymal lesion was seen in a single patient (25.0%). No spinal leptomeningeal, pachymeningeal, or cauda equina enhancement was seen in these cases.

^1^HRS-MRI was performed in a single patient (with a dominant lesion in the left putamen) and was found to be normal after treatment. Magnetic resonance perfusion imaging was not performed in any cases. 

In the reported series of Bou et al. [8], a minimum diameter of 1 cm was proposed as an inclusion criterion, smaller in comparison to the size of 2 cm typically used to define TDLs [11], but usually parenchymal lesions in neurosarcoidosis are less than 1 cm in size [134]. In addition, leptomeningeal contrast enhancement on standard 1.5 T MRIs is a common finding in neurosarcoidosis in general and especially in the setting of brain parenchymal lesions, being extremely rare in TDLs (MS in particular) [135]. Then, the authors proposed that brain parenchymal lesions in neurosarcoidosis be considered tumefactive when their largest diameter exceeds 1 cm and is associated with vasogenic edema and/or mass effect. Although there is a predilection of neurosarcoidosis for the posterior fossa, tumefactive parenchymal lesions [8,134] occur twice as commonly in the supratentorial brain as in the infratentorial compartment. The most affected regions were the frontal lobes, subinsular region, and basal ganglia. The morphology of the lesions could be spherical > stellate, and all lesions were contrast-enhancing, more commonly in a heterogeneous but well-demarcated fashion. Perilesional vasogenic edema was present and often significant in all cases, frequently contributing to the mass effect, mimicking malignancies and other mass-like lesions in the brain. 

### 3.6. Neurotoxoplasmosis

Neurotoxoplasmosis is one of the most frequent cerebral opportunistic infections in patients with HIV [136], involving nearly 40% of the HIV-infected population and over 75% on autopsy [137]. In addition, one third of patients with severe immunosuppression may develop cerebral toxoplasmosis within 12 months [138]. The initial diagnosis of neurotoxoplasmosis in HIV patients is usually carried out empirically if multiple ring-enhancing lesions are present on brain MRI, serology is positive for Toxoplasma gondii in the CSF or blood, and subsequent clinico-radiological improvement after anti-toxoplasma therapy is observed. Empiric diagnosis can be problematic because elevated serum IgG anti-toxoplasma titers are common in the general population in the absence of active disease [139,140,141]. 

Classical imaging features of neurotoxoplasmosis are multifocal ring-enhancing lesions in the cortex and periventricular white matter. Moreover, when present, an “eccentric target sign” consisting of an eccentric nodule along the rim of an enhancing lesion on T1-WI is considered pathognomonic [142] and has 95% specificity and less than 30% sensitivity [143]; it occurs in lesions on the brain surface. Another typical feature is a concentric target sign on T2-WI/FLAIR, described as a central hypointense lesion, a hyperintense intermediate region, with a hypointense halo visualized by edema around the lesion [144]. Notably, T1WI and T2WI features, if present together, are not usually associated with the same lesion. In a series of 27 patients, Brightbill et al. [144] focused on T2W- MRI findings with 10/27 (37%) mainly hyperintense lesions, 10/27 (37%) isointense, and 7/27 (26%) mixed lesions. The histopathological correlate of iso-hypointense lesions was the formation of abscesses. Restricted diffusion can sometimes be seen [145]. Tumefactive forms of neurotoxoplasmosis occur as single, large, necrotic lesions with extensive adjacent edema (sometimes called acute necrotizing toxoplasmosis), and they can be almost indistinguishable from lymphoma. In such tumefactive cases, advanced imaging features such as diffusion, perfusion, and spectroscopy can be sometimes useful to distinguish toxoplasmosis from lymphoma [56]. PCNSL is one of the main differential diagnoses of neurotoxoplasmosis, in particular in the tumefactive variant. In this regard, slightly higher relative apparent diffusion coefficient values are reported in toxoplasmosis (ADC > 1.6 × 10^−3^ mm^2^/s suggestive of toxoplasmosis; ADC < 1.1 × 10^−3^ mm^2^/s pointing to lymphoma; intermediate values reported for both), as well as lower rCBV values (rCBV > 1.5 more common in lymphoma) [146,147,148]. In the paper by Dibble et al. [148], previous studies used DSC-MRI for distinguishing abscess from glioblastoma and metastasis, and showed that the rCBV within the enhancing rims of abscesses is typically lower than that in necrotic glioblastomas and metastases [149,150]. A previous study [151] found significantly higher rCBV in tumors (4.28 ± 2.11) compared to infections (0.63 ± 0.49), with a threshold of 1.3 yielding a sensitivity of 97.8% and a specificity of 92.6%. Nearly all infections studied were toxoplasmosis (38/46), but PCNSL was not included among the neoplasms. In the study of Dibble et al. [148], the rCBV in neurotoxoplasmosis lesions (average mean lesion rCBV = 1.03, with some approaching 2) tended to be higher than those in the study by Floriano et al. [151] (0.63), likely due to methodological differences, including their use of spin-echo rather than gradient-echo DSC-MRI and the lack of post-processing leakage correction. Another reason for this finding may be the concurrent abscess presence in neurotoxoplasmosis lesions, because the newly formed capillaries in abscess capsules lack tight junctions and are therefore associated with increased vascular permeability [152], so contrast extravasation in toxoplasmosis may artifactually lower rCBV estimates unless corrected for.

A markedly increased choline peak on MR spectroscopy could also favor lymphoma over infection, but false positives are possible. 

In a single-center series [153] of 15 patients with neurotoxoplasmosis with a detailed analysis of neuroimaging, intralesional susceptibility signals (ISSs) on SWI were identified in 14/15 (93.3%) patients (mean size 5.2 ± 3.8 mm), with an average number of ISS 3.9 lesions/patient and 4/15 (26.7%) patients having a single ISS. The average number of abnormal foci per MRI sequence was FLAIR = 12.4 lesions/patient, CE T1WI = 8.2 lesions/patient, DWI = 7.1 lesions/patient, and non-contrast T1WI = 3.4 lesions/patient. On DWI, 13/15 (86.7%) patients had at least one hyperintense lesion with a hypointense signal on ADC maps. 

SWI could be useful in characterizing neurotoxoplasmosis, but it is not possible to differentiate, on this basis, neurotoxoplasmosis from PCNSL, in particular in immunocompromised patients.

The main features of MRI in differential diagnoses of tumefactive brain lesions are summarized in Table 4 and Table 5.

## 4. MRI as a Tool for Decision Making 

Tumefactive brain lesions are defined as mass-like or tumor-like lesions and tumors are the main mimics. The differential diagnosis of tumor-like brain lesions relies on clinical, neuroimaging, and pathological data in close integration. The contribution of neuroimaging, in particular conventional and advanced MRI techniques, is of paramount importance for guiding the diagnostic pathway. In this subset of patients, its contribution needs standardization because only the neuroimaging of tumors and, partially, TDLs, has already been agreed in detail. This is a crucial issue when other diseases are considered in differential diagnoses. The main purpose of this review is to provide an update on the neuroimaging patterns and parameters useful for supporting and orienting a differential diagnosis, strongly considering the contribution of advanced MRI techniques. 

Differentiation between t-PACNS and brain tumors (in particular HHG) on imaging alone can be challenging. Advanced imaging techniques, such as DSC-MRI and ^1^HSR-MRI, may support the diagnosis, because t-PACNS usually does not show increased rCBV, which is a feature of neoangiogenesis seen in HGG. Nevertheless, advanced MRI techniques have been rarely used for the imaging of t-PACNS; then, the actual data come from a very small subset of patients and a non-standardized investigation strategy. 

Differential diagnosis based on imaging findings would often include PCNSL, and intralesional microhemorrhages are not an uncommon imaging feature of PCNSL [154]. They therefore have no strong supporting value in the differential diagnosis between these two diseases, in isolation. However, systemic lymphoma can occur in about 6% of patients with PACNS [54], both as a prior history or simultaneous disease detected on PACNS work-up. From a neuroimaging point of view, the lack of diffusion restriction may be helpful in differentiating t-PACNS from PCNSL, in particular considering single lesions. It is difficult to differentiate between PACNS and PCNSL when there are multiple-enhancing lesions with vasogenic edema on MR imaging (e.g., case 2 and 4 in Lee et al.) [21], and sometimes follow-up neuroimaging might be helpful to remove some doubts. One of the main issues in this regard is the effect of steroid treatment on the neuroimaging evolution of the lesions in the case of PCNSL. Another issue is that sometimes histopathology shows a similarity of lymphoma and PACNS, particularly lymphocytic vasculitis, and it is still of paramount importance that there is clear documentation of transmural infiltration and not only a perivascular infiltrate to diagnose PACNS, in particular in immunocompetent patients where the PACNS-related mass lesion might simulate some features of well-enhancing lymphoma on MRI [155]. This issue should be strongly considered, in particular when the sampled lesions have a necrotic or hemorrhagic component with a secondary injury of the brain tissue. Moreover, in a small case series [21], DWI findings were highly variable among the patients, providing little aid in the diagnosis of PACNS.

Another neuroimaging issue helping in differential diagnosis is the pattern of contrast enhancement of the single or multiple lesions and of the other regions of the brain and leptomeninges. Unfortunately, the pattern of contrast enhancement may be variable in tumor-mimicking PACNS, ranging from irregular subcortical patchy [10,21] or streaky enhancement to tumor-like enhancement [21]. Moreover, small punctate, leptomeningeal, and focal cortical ribbon-like enhancement has been reported in non-tumor-mimicking PACNS and it may coexist with the contrast enhancement pattern of t-PACNS. Moreover, if the presence of intralesional microhemorrhages and the lack of specific open-ring enhancement or peripheral diffusion restriction assisted in differentiating t-PCNSV from TDLs, nevertheless, open-ring enhancement is present in only 35% of patients [89] with TDLs, making this marker less useful in routine clinical practice, in particular when it is not present. Both the tumefactive form of neurosarcoidosis and neurotoxoplasmosis can be easily differentiated from t-PACNS when they are not the presenting manifestations of the disease and when a typical clinical context is present. Otherwise, the differential diagnosis may be challenging, in particular for tumefactive neurosarcoidosis, presenting often with a single lesion and not always in locations considered highly suggestive for it. One of the cases described by Lee et al. [21] in the suprasellar area with low ADC values might mimic neurosacoidosis. 

Other unusual atypical infectious conditions such as CNS tuberculosis and fungal infections should also be included in the differential diagnosis of t-PCNSV. MRI techniques are often not applied in the published case series of patients without an initial hypothesis of primary brain tumor, so more data should be acquired to define the findings in t-PACNS in comparison with other tumefactive brain diseases. 

Sometimes, the clinical and radiological features are various and nonspecific, making correct diagnosis difficult without biopsy.

A further issue is represented by the choice of tumefactive brain diseases in differential diagnosis with tPACNS. In fact, the selected diseases do not cover the full range of differential diagnosis and some rare diseases in individual cases deserve a mention, for example Acute Hemorrhagic Leucoencephalitis (AHLE), also known as Weston–Hurst syndrome or Hurst disease, which is a “unicum” among the TDLs [155,156]. Their features are larger confluent lesions with significant edema and space-occupying effect in FLAIR-MRI and coexistent hemorrhages. The findings of advanced MRI techniques in these patients are unknown or not reported. These considerations apply to several different diseases, which were not included in this review. 

## 5. Conclusions

Tumefactive brain lesions are a complex issue and several diseases have to be considered in differential diagnosis. PACNS is a rare disease and t-PACNS is a rare manifestation of a rare disease, so differential diagnosis is challenging in most cases. Neuroimaging findings are one of the main elements contributing to differential diagnosis and orienting the diagnostic pathway. Among neuroimaging findings, advanced MRI techniques might help, in particular when a typical pattern attributed to primary brain tumors is present (HGG and PCNSL). The other diseases considered in this review, including PACNS, have a less standardized MRI protocol and advanced MRI techniques have not been systematically applied or reported, so less information is available. Starting from PACNS and t-PACNS, a standardization of a modern neuroimaging protocol, including advanced MRI, is lacking and it could help in the definition of clinical, neuroimaging, and pathological phenotypes and their correlations, making the differential diagnosis from other tumefactive brain diseases easier. 

## Figures and Tables

**Figure 1 diagnostics-14-00618-f001:**
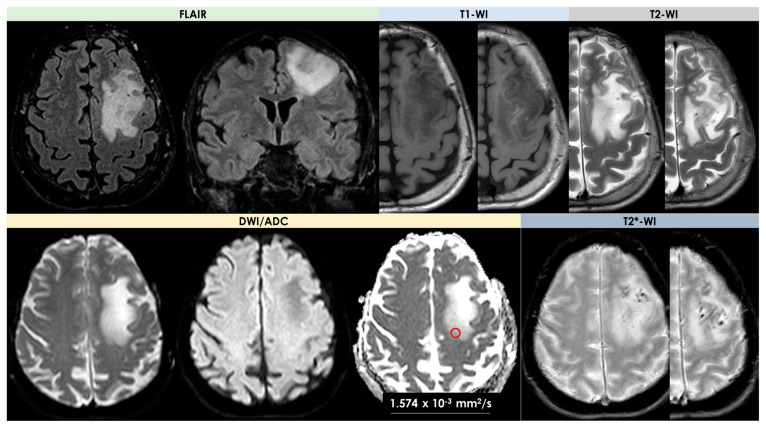
Conventional MRI findings in t-PACNS. A cortico-subcortical left frontal lesion is illustrated, hyperintense on T2/FLAIR (axial and coronal views) and T2WI and hypointense on T1-WI. DWI and ADC images show a corresponding hypo- and hysointense signal, respectively. In the GRE–T2* sequence, small punctate hypointense regions are evident within the lesion. ADC_min_ values are reported in the figure with the corresponding region of interest (ROI) (red circle).

**Figure 2 diagnostics-14-00618-f002:**
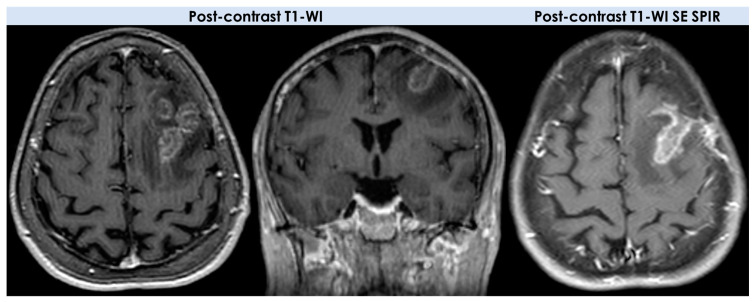
Post-contrast MRI sequences in the same patient depicted in Figure 1, showing a peripheral and mainly cortical rim of contrast enhancement.

**Figure 3 diagnostics-14-00618-f003:**
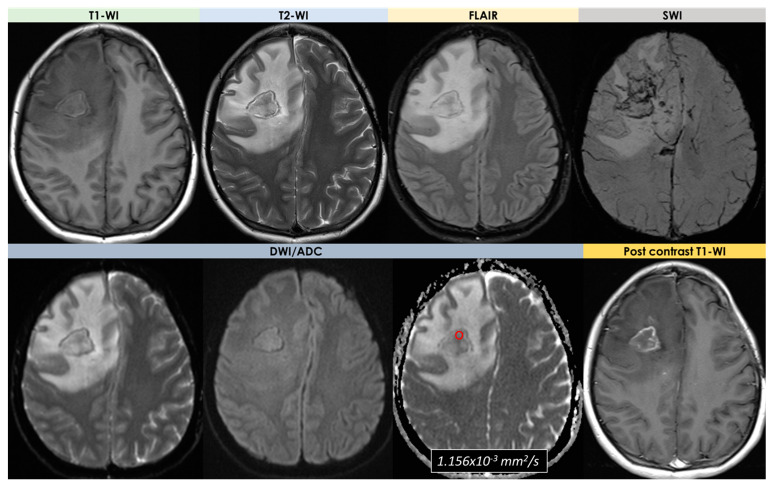
Another example of MRI findings in a patient with t-PACNS, showing a subcortical right frontal mass lesion with peripheral contrast enhancement surrounded by edema and including SWI hypointense foci. ADC_min_ values are reported in the figure with the corresponding ROI (red circle).

**Figure 4 diagnostics-14-00618-f004:**
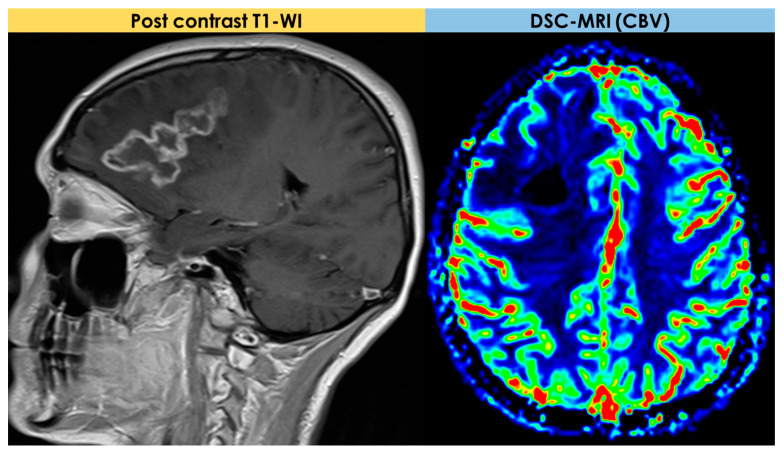
Advanced MRI technique findings in t-PACNS. The subcortical peripheral-enhancing right frontal lesion is characterized by low rCBV values on DSC-MRI.

**Figure 5 diagnostics-14-00618-f005:**
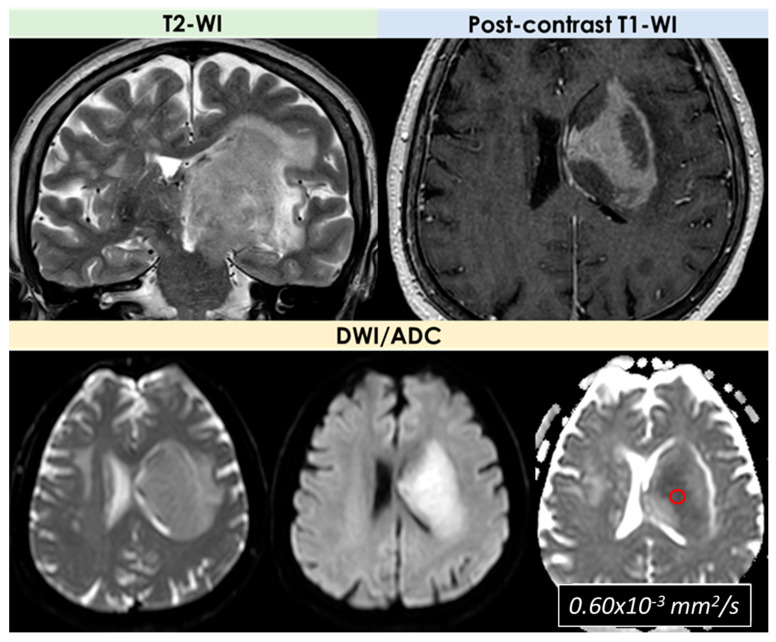
Conventional MRI findings in PCNSL. The T2-weighted imaging (T2WI) signal shows a mass lesion characterized by hyperintensity with mottled internal hypointense areas. On post-contrast T1-weighted imaging, a lesional enhancement is evident with some hypoenhancing areas within it. DWI with ADC shows a prevalent internal restriction. ADC_min_ values are reported in the figure with the corresponding region of interest (ROI) (red circle).

**Figure 6 diagnostics-14-00618-f006:**
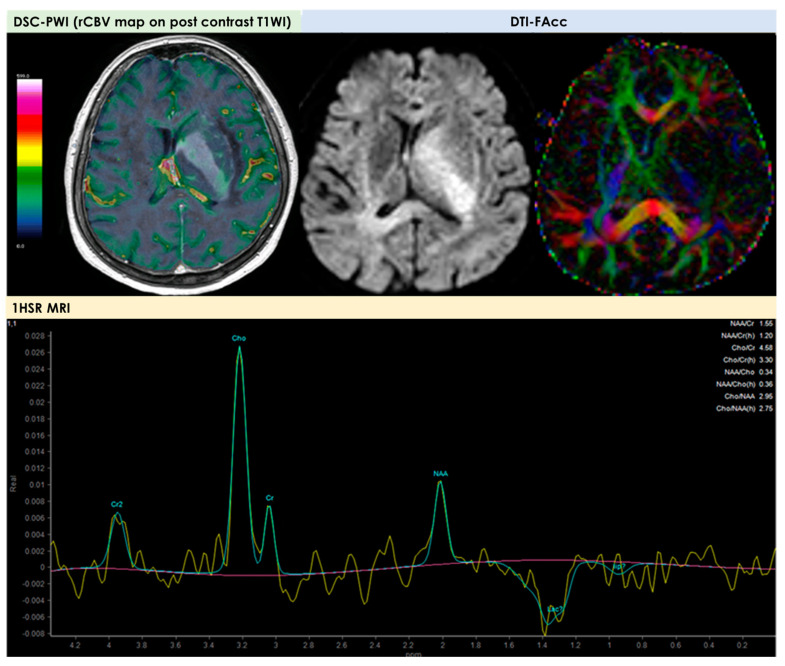
Advanced MRI findings in PCNSL. Dynamic susceptibility contrast (DSC) MRI measures of rCBV define microvascular volume as an indicator of tumor-related angiogenesis. Diffusion tensor imaging–fractional anisotropy color-coded (DTI-FAcc) shows the integrity of white matter tracts. ^1^HRS spectroscopy demonstrates a pathological increase in the Cho peak and a decrease in the NAA peak (Cho/Cr 3.30, NAA/Cho: 0.36 e Cho/NAA 2.75).

**Figure 7 diagnostics-14-00618-f007:**
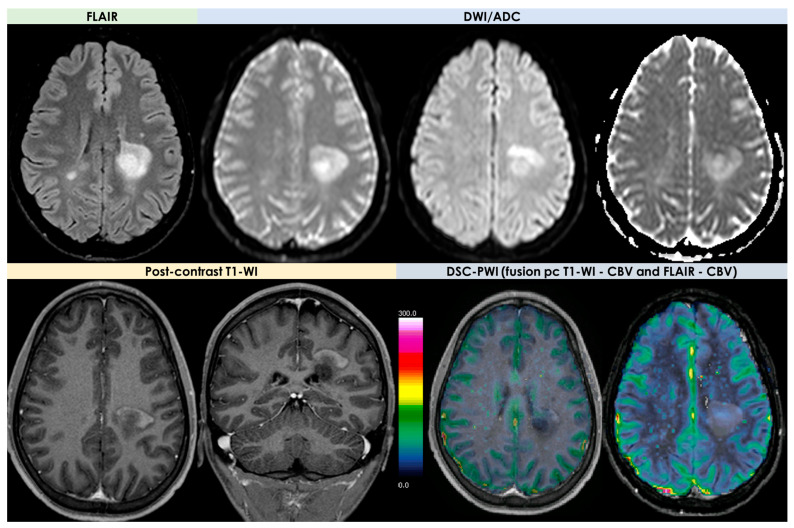
MRI pattern in MS-related TDL. A subcortical white matter heterogeneously hyperintense on T2/FLAIR lesion is shown, with a mild tumefactive effect on the ventricular wall. In DWI sequences, there is not a clear restriction. Post-GBCA T1W sequences show heterogenous enhancement with an incomplete ring pattern. DSC-MRI shows a hypoperfusion of the mass-like lesion.

**Figure 8 diagnostics-14-00618-f008:**
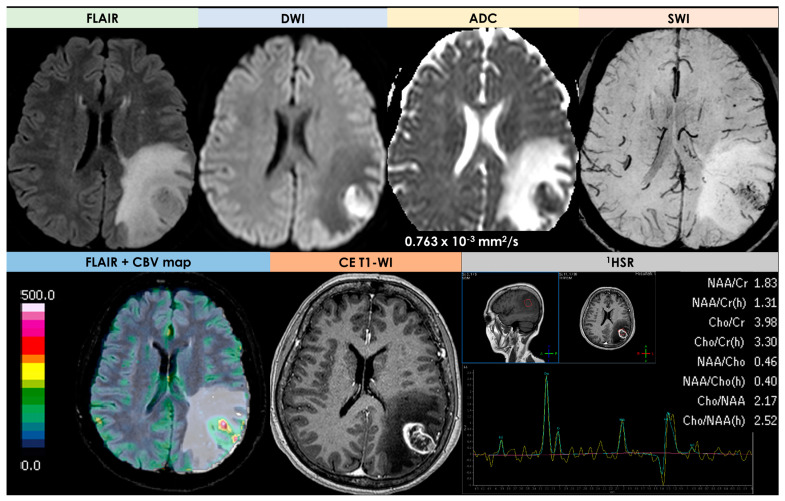
MRI pattern in HGG. The MRI shows a parietal lesion with mass effect and surrounding edema. The lesion is characterized by a heterogeneous signal with prevalent hyperintensity on T2-FLAIR with regressive necrotic areas on post-contrast T1W with vivid peripheral ring enhancement. DCS-MRI shows hyperperfusion with high rCBV values. ^1^HRS shows an elevation of the Cho peak with a high Cho/NAA ratio.

**Table 1 diagnostics-14-00618-t001:** Neuroimaging patterns of PACNS according to the ESO guidelines [5].

Neuroimaging Patterns	Reported Rate
Acute intracerebral hemorrhage (ICH)/subarachnoid hemorrhage (SAH)	90/660 (13.6%)
Tumefactive (or pseudotumoral) pattern (t-PACNS)	27/660 (4.1%)
Multiple acute/subacute ischemic lesions	42/660 (6.4%)
Single acute/subacute ischemic lesion	123/660 (18.6%)
Small vessel disease (SVD) pattern (according to the STRIVE criteria) [6,7]	58/660 (8.8%)
Parenchymal contrast enhancement	135/660 (20.4%)
Spinal cord involvement	5/660 (0.8%)

**Table 2 diagnostics-14-00618-t002:** MRI technical features in t-PACNS cohorts.

Cohort.	Suthiphosuwan et al. [10]	Lee et al. [21]	De Boysson et al. [16]	Salvarani et al. [13]
Field strength	1.5 T and 3.0 T	1.5-T	1.5 T and 3.0 T	Not reported
Standard MRI sequences	T1-WI, T2-WI, FLAIR, and DWI. GRE-T2* or SWI Gadolinium-enhanced T1WI	Spin-echo T1-WI, fast spin-echo T2-WI, contrast-enhanced T1-WI in multiple planes, DWI with b = 1000 s/mm^2^	Not detailed	Not reported
Vascular imaging	CTA or MRA in 9/10 pts	3D TOF MRA DSA (high resolution matrix 1024 × 1024) in three out of four patients	MRA in all patients *	Not fully reported * Angiogram showing vasculitis was present in one out of three of the t-PACNS patients
Advanced MRI	DSC-MRI (3/10) ^1^HRS-MRI (2/10) VWI (3/10)	DSC-MRI in 3/4 patients Single-voxel ^1^HSR- MRS in two out of four patients	Not detailed	Not reported

* FLAIR: Fluid-Attenuated Inversion Recovery; TOF: Time of Flight; DSA: Digital Subtraction Angiography; DSC: Dynamic susceptibility contrast; ^1^HSR: Proton Spectroscopy; VWI: Vessel Wall Imaging.

**Table 3 diagnostics-14-00618-t003:** Conventional MRI findings in TDLs with pooled incidence [90].

MRI Findings	Pooled Incidence (95% CI)
Open-ring or incomplete-rim enhancement	35% (24–47%)
Closed-ring or complete-rim enhancement	18% (11–29%)
T2 hypointense rim	48% (36–60%)
Absent or mild mass effect	67% (48–83%)
Absent or mild perilesional edema	57% (36–76%)

**Table 4 diagnostics-14-00618-t004:** MRI features of the main differential diagnoses of t-PACNS in their tumefactive forms.

MRI	PCNSL	TDLs	High Grade Glioma	Neurosarcoidosis	Toxoplasmosis
Preferential location	Periventricular and superficial brain regions	Mainly supratentorial (frontal and parietal lobe)	Subcortical white matter and deeper gray matter of the cerebral hemispheres, brainstem, cerebellum, and spinal cord	Single or multiple supratentorial lesions (spherical > stellate) with prominent perilesional edema	Usually multiple lesions, but the tumefactive form shows a single, large, necrotic lesion with extensive adjacent edema (acute necrotizing toxoplasmosis)
T1-WI	Hypo- or isointense	Heterogeneous with prevalent hypointensity	Hypointense	Heterogeneous with prevalent hypointensity	
T2-WI and FLAIR	Iso- or hypointense	Heterogeneous with hypointense rim (48%) Absent or mild mass effect (67%) Absent or mild perilesional edema (57%)	Hyperintense surrounding edema	Heterogeneous with central hypointensity Sometimes hiso-hypointensity	Concentric target sign on T2-WI/FLAIR (central hypointense lesion, hyperintense intermediate region, surrounded by a hypointense halo)
Contrast enhancement (CE) on T1-WI	Moderate–marked CE Non-AIDS patients: homogeneous CE, 90%; ring CE, 0–13% Leptomeningeal, subependymal, dural, or cranial nerve CE AIDS patients: irregular CE common; ring CE, 75%	Open ring (35%) Closed ring (18%) Homogeneous, heterogeneous, patchy and diffuse, cotton ball, and nodular patterns (NMOSD lesions often have absent or minimal enhancement)	Typically ring enhancement	Heterogeneous	

**Table 5 diagnostics-14-00618-t005:** Advanced MRI features of the main differential diagnoses of t-PACNS.

MRI	PCNSL	TDLs	High Grade Glioma	Neurosarcoidosis	Toxoplasmosis
DWI/ADC	Hyperintense on DWI/hypointense on ADC maps Decreased FA values in lesions	High ADC values in center of lesion and relatively low ADC values in periphery of lesion Threshold for the minimum ADC value for distinguishing TDLs from PCNSL is 0.556 × 10^−3^ mm^2^/s	Hypointense in DWI and ADC	Hypointense on DWI and ADC	“Eccentric target sign” consisting of an eccentric nodule along the rim of an enhancing lesion on T1-WI ADC > 1.6 × 10^−3^ mm^2^/s suggestive of toxoplasmosis
DSC-MRI	rCBV lower than HGG Characteristic TIC	Few data with variable findings	rCBV usually higher than PCNSL, but variable findings	No data	rCBV lower than that in necrotic glioblastomas and metastases
Spectroscopy	Elevated lipid peaks and high Cho/Cr ratios	Increased Cho- peak Decreased NAA peak Increased Cho/NAA ratio	Elevated lipid peaks High Cho/Cr ratio	No data	No data

## Data Availability

Not applicable.

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
