# Peer review of "Tumor-like Lesions in Primary Angiitis of the Central Nervous System: The Role of Magnetic Resonance Imaging in Differential Diagnosis"

_diagnostics, 2024, doi:10.3390/diagnostics14060618_

Round 1
Reviewer 1 Report
Comments and Suggestions for Authors
very well written review article
I would suggest to compare AHL (Hurst), which you mentioned, with tPACNS.
Also it would be very helpful, if you would instead or in addition to table 4 and 5 give a more simple table, just compare the main features of tPACNS and its differential diagnosis in T2, DWI, CBF/CBV, CE, spectroscopy ... e.g. with +++ signs or just the best clue. So everybody can have a clear take home message.
Author Response
First of all, we would like to thank the reviewer for his/her appreciation of our work and for his/her suggestions.
As you perfectly undelined, the topic is really challenging and the neuroradiological features of the diseases described in the text are sometimes atypical, variable and not so easy to summarize. In our first attempt, we tried to draft a more simple table, as suggested, but the findings were not satisfying, because of the lack of strong information on advanced MRI in these patients and because of the the frequent presence of atypical forms.
We added a sentence on AHL. It is an interesting proposal, but, unfortunately, information about advanced MRI in these patients is lacking, so a formal comparison is really complicated.
"
A further issue is represented by the choice of tumefactive brain diseases in differential diagnosis with tPACNS. In fact, the selected diseases do not cover the full range of differential diagnosis and some rare diseases in individual cases deserve a mention, for example Acute Hemorrhagic Leucoencephalitis (AHLE), also known as Weston-Hurst syndrome or Hurst disease, which is a “unicum” among the TDLs [161,162]. Their features are larger confluent lesions with significant edema and space-occupying effect in FLAIR-MRI and coexistent hemorrhages. The findings of advanced MRI techniques in these patients are unknown or not reported. These considerations apply to several different diseases, which were not included in this review. "
Reviewer 2 Report
Comments and Suggestions for Authors
The authors presentend a comprehensive overview of differential diagnosis of primary angiitis of the central nervous system which is a rare neurological disease.
I find this paper very usefull for daily neuroradiological practice.
Author Response
Many thanks for your appreciation of our paper!